# Evaluation of Anti-Biofilm Activity of Mouthrinses Containing Tannic Acid or Chitosan on Dentin In Situ

**DOI:** 10.3390/molecules26051351

**Published:** 2021-03-03

**Authors:** Anton Schestakow, Moritz S. Guth, Tobias A. Eisenmenger, Matthias Hannig

**Affiliations:** Clinic of Operative Dentistry, Periodontology and Preventive Dentistry, University Hospital, Saarland University, Building 73, 66421 Homburg/Saar, Germany; moritzguth@gmx.de (M.S.G.); ft.eisenmenger@web.de (T.A.E.); matthias.hannig@uks.eu (M.H.)

**Keywords:** tannic acid, chitosan, preventive dentistry, biofilm

## Abstract

In contrast to enamel, dentin surfaces have been rarely used as substrates for studies evaluating the effects of experimental rinsing solutions on oral biofilm formation. The aim of the present in situ study was to investigate the effects of tannic acid and chitosan on 48-h biofilm formation on dentin surfaces. Biofilm was formed intraorally on dentin specimens, while six subjects rinsed with experimental solutions containing tannic acid, chitosan and water as negative or chlorhexidine as positive control. After 48 h of biofilm formation, specimens were evaluated for biofilm coverage and for viability of bacteria by fluorescence and scanning electron microscopy. In addition, saliva samples were collected after rinsing and analyzed by fluorescence (five subjects) and transmission electron microscopy (two subjects) in order to investigate the antibacterial effect on bacteria in a planktonic state and to visualize effects of the rinsing agents on salivary proteins. After rinsing with water, dentin specimens were covered by a multiple-layered biofilm with predominantly vital bacteria. In contrast, chlorhexidine led to dentin surfaces covered only by few and avital bacteria. By rinsing with tannic acid both strong anti-adherent and antibacterial effects were observed, but the effects declined in a time-dependent manner. Transmission electron micrographs of salivary samples indicated that aggregation of proteins and bacteria might explain the antiadhesion effects of tannic acid. Chitosan showed antibacterial effects on bacteria in saliva, while biofilm viability was only slightly reduced and no effects on bacterial adherence on dentin were observed, despite proteins being aggregated in saliva after rinsing with chitosan. Tannic acid is a promising anti-biofilm agent even on dentin surfaces, while rinsing with chitosan could not sufficiently prevent biofilm formation on dentin.

## 1. Introduction

Dental caries is the most prevalent chronic disease worldwide [1]. Even though caries is rarely life threatening, operative treatment of carious lesions is expensive and untreated caries can have further impact on oral health [2,3]. Therefore, the trend is moving towards prevention now and, since caries is a multifactorial disease, there are several targets where prophylaxis can take effect [4].

Caries is defined as the destruction of dental hard tissues due to acids made by bacteria within a biofilm when frequently exposed to sugars [4]. Hence, biofilm control, mechanically or chemically, is an important part of caries prophylaxis [5].

For mechanical biofilm control, teeth brushing with fluoridated toothpaste has been established. Both teeth brushing at high frequency and usage of higher fluoride concentrations can reduce the incidence of caries [5,6]. It is not necessary to change established preventive measurements such as teeth brushing twice a day with a fluoridated toothpaste. However, mechanical biofilm control requires compliance and manual skills, and consequently, it is difficult for incompliant people and mentally or physically impaired people to maintain high oral hygiene standards with mechanical biofilm control alone [7]. In this case, chemical biofilm control can meet the demands.

For chemical biofilm control, chlorhexidine (CHX) is considered the gold standard [8]. CHX can prevent biofilm formation by a prolonged antibacterial and anti-adherent effect, which is the result of CHX’s retention in the oral cavity [9,10,11]. However, side effects such as tooth discoloration, taste irritation and irritation of the oral mucosa limit the use of CHX [12]. Some alternates to CHX have already been investigated [13], of which natural products have become more and more popular [14].

Natural products have always been of importance for drug discovery, and they have been used to treat oral diseases for thousands of years [15,16]. They can inhibit caries progression through an antibacterial or anti-adherent effect or by inhibition of the polysaccharide synthesis. Due to technological advances, it has also become easier to identify and examine active components of natural products [17], which are often polyphenols [18]. In the present study, the natural substances tannic acid and chitosan were investigated.

Tannic acid belongs to the tannins, a group of water-soluble polyphenols that are often found in plants, protecting them from herbivores and decay. Therefore, it is not surprising that oak or chestnut trees, which are known for their high durability, are rich in tannins [19,20]. Due to astringent properties, tannins can not only inhibit proteins that are necessary for bacterial adherence [21], they also have an antibacterial effect through their chelating properties [22]. In addition, tannins may also interact with membranes, leading to leakage of internal contents or even to bursting of cells [23]. With regard to dental biofilm formation, several in situ studies have already investigated the effects of tannins showing anti-adherent, antibacterial and anti-erosive properties of tannic acid in particular [24,25].

In contrast to tannic acid, the other test substance, chitosan, is a semi-synthetic material made from chitin [26]. Chitin is a very common natural polymer and can be obtained from fungi, especially from arthropods [27]. Since chitin is highly insoluble, the soluble deacetylated derivate chitosan is used for practical applications [28]. Chitosan has antibacterial properties thanks to its positively charged amino groups in acidic aqueous solutions [29]. In the literature, chitosan is often used in different application forms or derivatives [26,30]. Nevertheless, mouthrinses containing water-soluble chitosan can reduce plaque indices as well as bacterial viability [31,32].

The present study is a follow-up of the previously published study by Schestakow and Hannig [33] regarding the effects of tannic acid and chitosan on biofilm formation on enamel in situ. Since dentin exposure to the oral cavity is more common as the population ages [34], the anti-adherent and antibacterial effect of tannic acid and chitosan was further investigated on biofilm formation on dentin. For this purpose, bovine dentin specimens were placed in the oral cavity for 48 h to enable biofilm formation while subjects rinsed with solutions containing tannic acid or chitosan as well as water or CHX as controls. Two rinsing protocols were applied, which differed in the number of rinses and thus the time interval between the last rinse and the ex vivo examination allowing the investigation of immediate and long-term effects by fluorescence microscopy (FM) and scanning electron microscopy (SEM). In order to investigate the mode of action with regard to anti-adherent and antibacterial properties, the interaction of test substances with non-adherent bacteria in saliva was examined by using FM and transmission electron microscopy (TEM).

## 2. Results

### 2.1. FM Analysis of the Biofilm

When subjects rinsed with the negative control water, specimens were predominantly covered by a multiple-layered biofilm regardless of the rinsing protocol (about 4.6) (Figure 1, Appendix A). Bacteria were mainly cocci; rods were less common. After rinsing with tannic acid according to rinsing protocol 1, specimens were significantly less covered by bacteria (2.8 ± 1), which were scattered or aggregated. Using rinsing protocol 2, specimens were more covered by biofilm (3.8 ± 0.8) after rinsing with tannic acid compared to applying rinsing protocol 1. After rinsing with chitosan, specimens were covered by a multiple-layered biofilm, which did not differ significantly from the negative control, regardless of the rinsing protocol (about 4.4). Both when using rinsing protocol 1 (1.9 ± 1) and rinsing protocol 2 (1.4 ± 0.3), after rinsing with CHX, specimens were only covered by scattered bacteria or small bacterial aggregations. The biofilm coverage was significantly reduced by rinsing with CHX according to both rinsing protocols compared to the negative control.

Bacteria in the biofilm were also evaluated for viability in order to investigate antibacterial properties of test substances. After rinsing with water according to protocol 1 (4.2 ± 0.5) and 2 (4.4 ± 0.6), most bacteria were vital in the biofilm (Figure 2). By rinsing with tannic acid applying protocol 1, the viability was reduced so that the biofilm contained vital and avital bacteria in equal amounts (2.9 ± 0.5). However, the biofilm contained more vital bacteria after rinsing with tannic acid according to protocol 2 (3.4 ± 0.2). The same applies to rinsing with chitosan (3.4 ± 0.6 and 4 ± 0.3). For the positive control CHX, however, the biofilm consisted mainly of avital bacteria, regardless of the rinsing protocol (about 2). The viability was significantly reduced in both rinsing protocols. The full dataset of the biofilm analysis can be seen in the Appendix A.

### 2.2. SEM Analysis of the Biofilm

Considering the results of FEM analyses, test substances have anti-adherent effects on dental biofilm formation in situ. In order to examine how test substances exert their anti-adherent effects, SEM was applied to investigate bacterial adherence under the influence of test substances. After 48 h of biofilm formation and four or five rinses, specimens were either covered by the pellicle that appeared as a layer of globular aggregates 100–200 nm in size, or by bacteria, which were mainly cocci and a few rods (Figure 3). After rinsing with water, tannic acid or chitosan, bacteria had an intact morphology with a globular structured surface representing the glycocalyx or the pellicle covering bacteria. Some bacteria had fimbriae that were linked to other bacteria, the biofilm matrix or the pellicle. In comparison to rinsing with water or chitosan, specimens were less covered by bacteria when subjects were rinsed with tannic acid. After rinsing with CHX, specimens were predominantly covered by bacteria-free pellicles with an altered structure consisting of globular agglomerates with a size of 200–500 nm. The few bacteria were isolated or in colonies. As a result, dentinal tubules were visible more frequently than after rinsing with water, chitosan or tannic acid, which contained thicker biofilms with characteristic water channels. Adherent bacteria also appeared in dentinal tubules.

### 2.3. FM Analysis of Saliva Samples

In order to further clarify the antibacterial effects of tannic acid, chitosan or CHX, as observed on biofilms by FM, saliva samples with non-adherent bacteria were investigated for viability after rinsing with different test substances. After rinsing with the negative control water, the viability of bacteria was over 70%, regardless of whether saliva samples were collected 1 min, 30 min or 2 h after rinsing (Figure 4, Appendix A). Viability of the salivary bacteria was reduced 1 min after rinsing with tannic acid (47 ± 6). The antibacterial effect declined after 30 min (64 ± 22) and reached the value of the negative control after 2 h (75 ± 13). The same applies to rinsing with chitosan (42 ± 17 and 61 ± 14 and 75 ± 9) with the antibacterial effect of chitosan being slightly stronger than of tannic acid. One min (33 ± 8) and 30 min (33 ± 8) after rinsing with CHX, bacteria were predominantly avital. After 2 h, about half of the bacteria were vital again (53 ± 12). In comparison to the negative control, CHX significantly reduced the viability after 1 min, 30 min and 2 h. The full dataset of the saliva analysis can be seen in the Appendix A.

### 2.4. TEM Analysis of Saliva Samples

An antibacterial effect of tannic acid, chitosan and CHX on non-adherent bacteria in saliva was examined quantitatively by FM. In comparison to FM, ultrastructural alterations of both bacteria and saliva can be visualized with TEM and, therefore, TEM was used to clarify the mechanism of action. One min, 30 min and 2 h after rinsing with the negative control water, intact bacteria were present, which were mainly cocci and a few rods with fimbriae covering the bacterial surface (Figure 5). Cleavage furrows were also visible, suggesting that bacteria were undergoing cell division at the time of fixation. In addition to bacteria, loose filamentous structures were detected representing salivary proteins. The bacteria were often found close to proteins or adsorbed to epithelial cells. When subjects rinsed with tannic acid, globular electron-dense structures appeared, which is the result of protein aggregation or formation of tannic acid-protein complexes. In particular, 1 min after rinsing with tannic acid, irregular shapes of bacteria were detected in contrast to the round and plump shape representing intact morphology. Furthermore, the bacterial cell wall had a higher electron density. Similar to tannic acid, protein aggregates were also visible with rinsing solutions containing chitosan, especially 1 min after rinsing. However, proteins were not aggregated into dense clusters as they were with tannic acid. Occasionally, cell remnants of bacteria were found within the protein networks. When subjects rinsed with the positive control CHX, cell lysis was frequently observed 1 min and 30 min after rinsing. CHX agglomerates appeared as globular electron-dense structures predominantly adsorbing and covering bacterial surfaces.

## 3. Discussion

Considering fluorescence microscopic analysis of the intraorally formed biofilm, rinsing solutions containing tannic acid, chitosan or CHX have an anti-adherent and antibacterial effect compared to the negative control water, with CHX showing the strongest and chitosan the weakest anti-biofilm effect. The antibacterial effect is the result of a disruption of membrane integrity, as shown by TEM. Furthermore, after rinsing with tannic acid, chitosan or CHX, ultrastructural alterations appeared in terms of protein aggregations and complexes that were also shown by SEM when rinsing with CHX caused alterations of the pellicle structure.

In the present study, tannic acid and native chitosan were tested for their anti-adherent and antibacterial effect on dental biofilm formation, since chitosan and polyphenols in general have often been the subject of medical research.

Tannic acid belongs to tannins, which are a subgroup of polyphenols that are known for their antibacterial activity due to chelating properties or interactions with the bacterial cell membrane when applied in high concentrations [22,35]. The antibacterial effect of tannic acid in particular was already investigated on dental biofilm formation. However, biofilm was either formed on enamel or biofilm formation time was low [24,25,33]. Since dentin is increasingly exposed to the oral cavity due to the aging population and the decline in edentulous adults [34], dentin specimens were used in the present study. When subjects rinsed with tannic acid, the viability of both non-adherent bacteria in the planktonic state and bacteria in biofilm was reduced. In order to investigate the duration of action, saliva samples were collected 1 min, 30 min and 2 h after rinsing with experimental solutions and for biofilm formation experiments two different rinsing protocols were applied. According to rinsing protocol 1, the last rinse was shortly before the ex vivo examination, and thus, the immediate effect on biofilm was investigated. For rinsing protocol 2, the last rinse occurred 12 h prior to the ex vivo examination, and thus, the long-term effect was examined. In view of this, the antibacterial effect decreased in a time-dependent manner, indicating poor retention of tannic acid in the oral cavity. Considering TEM analyses of saliva, tannic acid led to alterations of bacterial morphology, and thus, the antibacterial effect may be due to interaction of tannins with the bacterial membrane, as suggested by Tamba et al. [23], resulting in osmotic dysregulation and finally in cell death.

In addition to the morphological alterations, the formation of globular and electron-dense protein aggregations in saliva was observed by TEM, which occurred after 1 min and 30 min after rinsing with tannic acid. Since salivary proteins are used as receptors for initial bacterial adherence, precipitation of those proteins with the so-called tanning effect can explain the anti-adherent properties of tannic acid [36,37,38]. Tannins can also inhibit glycosyl-transferase, and thereby, glucan synthesis, which is used for bacterial adherence [39]. The same applies to bacterial fimbriae with which tannins can interact, as reported by Sakanaka et al. [40]. Furthermore, aggregations of bacteria can be observed in the presence of tannins [41], and therefore, bacteria may no longer adsorb to oral surfaces and are swallowed instead. Regarding the anti-adherent properties of tannic acid in particular, so far, one study showed an effect on biofilm formation on dentin that was exposed to the oral cavity [25]. In a study by Xi et al. [25], participants rinsed with a solution containing tannic acid (1%) twice a day; the biofilm was formed for 24 h. In the present study, when subjects rinsed with tannic acid (5%) and the biofilm formation time was 48 h, the anti-adherent effect of tannic acid was confirmed. However, 12 h after the last rinsing, as simulated by rinsing protocol 2, the effect was lower than in rinsing protocol 1, indicating a low substantivity of tannic acid.

In summary, rinsing with experimental solutions containing tannic acid inhibits biofilm formation. However, it is unclear to what extent tannic acid can disrupt an established biofilm. Furthermore, the exact mechanism of action of antibacterial effects of tannic acid in particular is still not fully clarified and should be further investigated.

The other test substance, chitosan, has been supposed as a promising anti-biofilm agent according to the literature [31,42,43,44,45,46], in which different chitosan derivatives or different application forms were investigated. However, the effect of native chitosan on intraoral biofilm formation on dentin has not been investigated yet. Chitosan has antibacterial properties due to positively charged groups. As a result, chitosan can disrupt the membrane integrity of bacteria or chelate metal ions [29,47]. Although chitosan did not lead to visible membrane interactions in the present study as shown by TEM, an antibacterial effect on non-adherent bacteria in saliva was observed especially 1 min after subjects rinsed with chitosan. The same applies to bacteria in the biofilm, but compared to the negative control, the antibacterial effect was very low and was only present in rinsing protocol 1. Applying rinsing protocol 2, on the other hand, no antibacterial effects were observed, which is in accordance with the results on bacteria in saliva 30 min and 2 h after rinsing with chitosan speaking for a low retention of chitosan in the oral cavity. Although a previous study on enamel led to similar results [33], the short duration of action of chitosan was not expected, since chitosan can adsorb to both buccal cells and the dental pellicle [46,48]. There are several factors that may lead to the limited activity of chitosan. The low pH value needed to dissolve chitosan and chitosan itself impart a positive charge to the pellicle in vitro [49,50] and, as suggested by Rehage et al. [51], this observation may inhibit further the accumulation of chitosan on dental surfaces. Furthermore, the salivary protein lysozyme, which is present in the pellicle maintaining its enzymatic activity, can degrade chitosan, and thus, inhibit its antibiofilm properties [52,53,54,55].

Considering transmission electron micrographs in the present study, rinsing agents containing chitosan led to the aggregation of proteins and bacteria due to the polycationic nature of chitosan [29,50]. Although the treatment of the pellicle with chitosan showed an anti-adherent effect in vitro [46,50], the anti-adherent properties of chitosan were not confirmed in the present in situ study, which is in accordance with a previous in situ study on enamel [33].

In contrast to tannic acid and chitosan, rinsing with the positive control CHX resulted in significant anti-adherent and antibacterial effects on biofilm formation on dentin regardless of the rinsing protocol as well as significant reduction of viability of non-adherent bacteria in saliva 1 min, 30 min and 2 h after rinsing with CHX. As a polycation, CHX can interact with the bacterial membrane, and thus, disrupt membrane integrity and cell metabolism [12]. The antibacterial effects were only observed by FM, but neither by SEM nor TEM. In the in vitro study by Vitkov et al. [56] when CHX was applied to saliva samples for 1 or 5 min, a loss of bacterial membrane integrity was visualized. In the present in situ study, however, no alterations occurred as shown by TEM, which may be due to the low concentration or the short rinsing time of only 30 s.

Unlike the other rinsing agents tested, alterations of the pellicle structure were detected by SEM. After rinsing with CHX, globular agglomerates with a size of 200–500 nm appeared. Since the polycation CHX can bind to negatively charged groups of salivary proteins, it is suggested that these agglomerates represent chlorhexidine-protein complexes in the pellicle leading to reduction of bacterial adherence [12,57]. In addition to adsorption to the pellicle, CHX also absorbs to other oral surfaces resulting in a high substantivity of CHX in the oral cavity [9,10,11]. When assessing significance, primarily CHX showed significant results. According to G*Power software, at least 12 subjects would be required to detect an 80% reduction in biofilm coverage or viability with a power of 80%. In the present study, six subjects participated. Subjects would be easier to hire when the number of rinsing solutions is reduced. The aim of the present study was also to show which substances actually work on dentin specimens. With the new findings, follow-up experiments concentrating specifically on one solution with a higher number of participants can be carried out.

## 4. Materials and Methods

### 4.1. Subjects and Test Substances

Six volunteers (aged 24–30 years) participated in the present study, which uses a cross-over design. All subjects were dental students who neither had caries nor periodontal diseases; they did not smoke nor take any drugs. The study was approved by the Medical Ethic Committee of the Medical Association of Saarland (238/03, 2016).

Subjects rinsed with four different mouth rinses. The washout phase was at least one whole day for all experiments, according to the substantivity of the positive control and the previous study on enamel [12,33,58]. Sterile water (Ampuwa^®^, Fresenius Kabi, Bad Homburg, Germany) and chlorhexidine-digluconat (0.2%) (Apotheke des Universitätsklinikum des Saarlandes, Homburg, Germany) were used as negative and positive control. Both tannic acid (Tannic Acid, Sigma^®^, Saint Louis, USA) and chitosan (Chitosan 95/3000, Heppe Medical Chitosan GmbH, Halle, Germany) were solids and had to be dissolved first. For 100 mL of a tannic acid solution (5%), sterile water was added to 5 g of tannic acid. To dissolve chitosan, sterile water was added to 5 g of chitosan and 3.5 mL of acetic acid to get a 1000 mL solution (0.5%). The chitosan used had a degree of deacetylation of ≥ 92.6% and a molecular weight of 300–700 kDa.

### 4.2. Specimens for Biofilm Formation

To investigate the effect on biofilm formation, six subjects carried upper jaw splints (DURAN^®^, Scheu Dental GmbH, Iserlohn, Deutschland) with dentin specimens that were fixed buccally with silicone impression material (PRESIDENT light body, Coltène/Whaledent GmbH + Co. KG, Langenau, Germany). Dentin specimens were made from bovine teeth from two-year-old cattle from the slaughterhouse in Zweibrücken by using a cut-off and wet grinding machine. They had a rectangular form with a surface of 5 × 5 mm^2^ and thickness of 1 mm and were ground and polished up to 2500 grit. The superficial smear layer was removed by ultrasonication with NaOCl (3%) for 30 s. Then, specimens were cleaned with distilled water and disinfected with isopropyl alcohol (70%) for 15 min before rehydrating in sterile water for 6 h [25].

### 4.3. Biofilm Formation In Situ

Four specimens were fixed to the splints, which were carried in the oral cavity to allow biofilm formation. Splints were in situ for 48 h, since biofilm thickness and viability are less susceptible to intraindividual differences in terms of the location of specimens [59,60]. Subjects rinsed four or five times with 10 mL of the different test substances for 30 s, as generally recommended for dental prophylaxis [61]. Two rinsing protocols were applied. In rinsing protocol 1, rinsing occurred 3 min, 12 h, 24 h, 36 h and 47.5 h after insertion of the splints. The last rinse was shortly before the ex vivo examination. In rinsing protocol 2, subjects rinsed only after 3 min, 12 h, 24 h and 36 h. During the trial, subjects had to temporarily take off the splints when they wanted to eat or brush their teeth, but usage of toothpaste or other mouth rinses were not allowed. After 48 h, splints were removed from the oral cavity and specimens were dismounted and rinsed with sterile water in order to remove non-adherent bacteria and salivary remnants. Specimens were then prepared for FM and SEM (Figure 6).

### 4.4. FM Analysis of the Biofilm

Two of the four specimens were stained with LIVE/DEAD^®^ BacLight™ Bacterial Viability Kit L7012 (Invitrogen, Molecular Probes, Eugene, OR, USA) for 10 min and then examined with FM (Axio Imager.M2, CarlZeiss Microscopy GmbH, Jena, Deutschland) using a fluorescein diacetate (Sigma, St. Louis, MO, USA) and an ethidium bromide filter (Roth, Mannheim, Deutschland) [33]. Six pictures of each specimen were taken, which were evaluated by two investigators for coverage and viability using a scoring system (Table 1 and Table 2).

### 4.5. SEM Analysis of the Biofilm

The other two specimens were prepared for SEM. First, specimens were fixed in a solution consisting of 2% glutaraldehyde and 0.1 M cacodylate buffer for at least 1 h. Then, specimens were washed in cacodylate buffer, dehydrated in an ascending alcohol series and dried with hexamethyldisilazane. After air drying overnight, specimens’ surface was coated with carbon and examined for its morphology with a magnification of up to 20,000 using SEM (XL 30 ESEM FEG, FEI Company, Eindhoven, The Netherlands).

### 4.6. FM Analysis of Saliva Samples

Five subjects rinsed with 10 mL of the different test substances for 30 s, and the unstimulated saliva was collected after 1 min, 30 min and 2 h in an Eppendorf tube. Samples were centrifuged for 10 min at 1000 rpm and the supernatant was centrifuged again for 10 min at 10,000 rpm. The bacterial pellet was stained with LIVE/DEAD^®^ BacLight™ for 15 min and examined with FM. Eight pictures were taken, and the viability of bacteria was evaluated using the software ImageJ 1.52 (NIH, Bethesda, MD, USA).

### 4.7. TEM Analysis of Saliva Samples

In order to visualize the interaction of test substances with bacteria, the saliva of two subjects was additionally examined with TEM. The subjects rinsed for 30 s with 10 mL of a test substance and their unstimulated saliva was collected after 1 min, 30 min and 2 h in an Eppendorf tube. The samples were centrifuged at 5000 rpm and the bacterial pellet was fixed in a fixing solution consisting of 1% formaldehyde, 1% glutaraldehyde and 0.1 M cacodylate buffer for 90 min. Then, samples were postfixed with 2% osmium for 1 h and pre-embedded in low-melting agarose. After dehydration in an ascending alcohol series, samples were embedded in araldite (Araldit CY212, Agar Scientific Ltd., Stansted, UK). Ultrathin sections of the embedded samples were cut in an ultramicrotome (Leica EM UC7, Leica Microsystems, Wetzlar, Germany). The sections were contrasted with UranyLess (UranyLess EM Stain, Delta Microscopies, Mauressac, France) and 3% lead citrate before investigated by transmission electron microscopy (TEM Tecnai 12 BioTwin, FEI Company, Eindhoven, Netherlands) at magnifications of up to 68,000-fold.

### 4.8. Statistical Analysis

The results of FM analyses were tested statistically. First, data were examined for normal distribution using the Shapiro-Wilk test. They were not normally distributed (*p* < 0.05). Statistical differences of the test substances to the negative control were tested with the Friedmann test (*p* = 0.05) followed by Dunn’s multiple comparison test. Differences between both rinsing protocols were tested with the Wilcoxon test (one-tailed). Bonferroni adjustments were conducted (*p* = 0.05/4 = 0.0125). Statistical analyses were performed with the GraphPad Prism 8 software (GraphPad Software, San Diego, CA, USA).

## 5. Conclusions

In conclusion, rinsing agents containing tannic acid reduced the bacterial viability and adherence to dentin specimens in situ due to interactions with bacterial membranes and proteins. Therefore, tannic acid is a promising anti-biofilm agent. On the other hand, rinsing with chitosan resulted in antibacterial effects on non-adherent bacteria in saliva and bacteria in the biofilm, but the antibacterial effect on biofilm formation was low and no anti-adherent properties were observed.

## Figures and Tables

**Figure 1 molecules-26-01351-f001:**
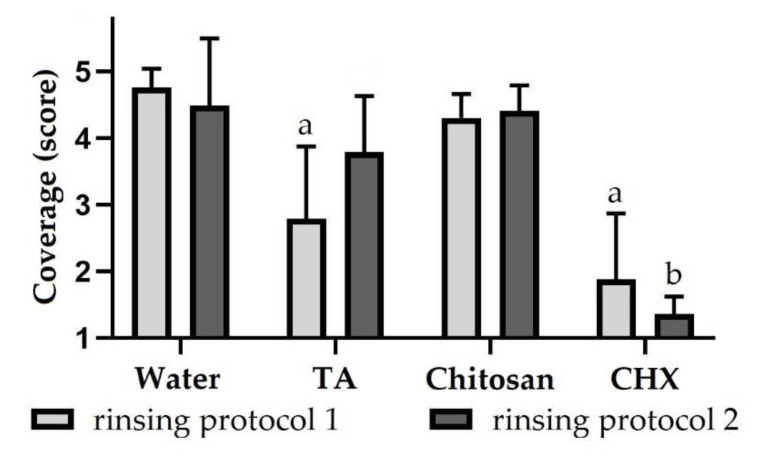
Biofilm coverage (score 1–5) of dentin specimens after rinsing with different rinsing solutions. Subjects carried intraoral splints with specimens for 48 h and rinsed with different experimental solutions. In rinsing protocol 1, rinsing was performed after 3 min, 12 h, 24 h, 36 h and 47.5 h and in rinsing protocol 2 after 3 min, 12 h, 24 h and 36 h. The biofilm formed on specimens was stained with LIVE/DEAD^®^ Baclight™ and evaluated using a scoring system. The height of the bars corresponds to mean values and the line applied to standard deviations. Friedman test followed by Dunn’s multiple comparison test: significant differences (*p* < 0.05) to water are marked with ‘a’ for rinsing protocol 1 and with ‘b’ for rinsing protocol 2. TA = tannic acid, CHX = chlorhexidine.

**Figure 2 molecules-26-01351-f002:**
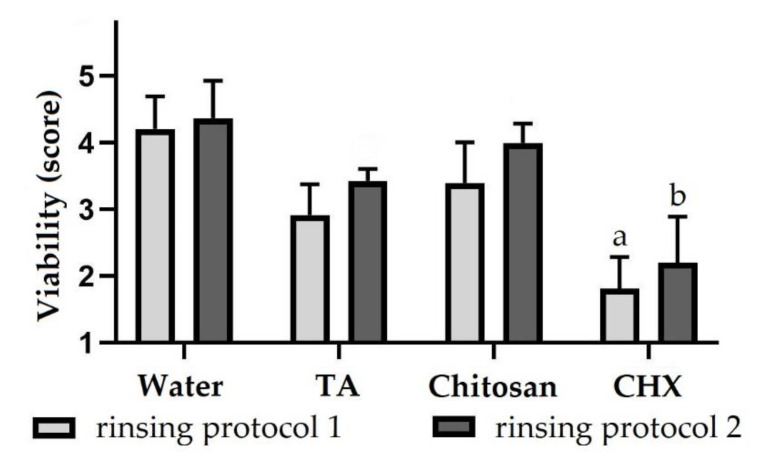
Bacterial viability (score 1–5) of biofilms formed on dentin specimens after rinsing with different rinsing solutions. Subjects carried intraoral splints with specimens for 48 h and rinsed with different experimental solutions. In rinsing protocol 1, rinsing was performed after 3 min, 12 h, 24 h, 36 h and 47.5 h and in rinsing protocol 2 after 3 min, 12 h, 24 h and 36 h. The biofilm formed on specimens was stained with LIVE/DEAD^®^ Baclight™ and evaluated using a scoring system. The height of the bars corresponds to mean values and the line applied to standard deviations. Friedman test followed by Dunn’s multiple comparison test: significant differences (*p* < 0.05) to water are marked with a for rinsing protocol 1 and with b for rinsing protocol 2. TA = tannic acid, CHX = chlorhexidine.

**Figure 3 molecules-26-01351-f003:**
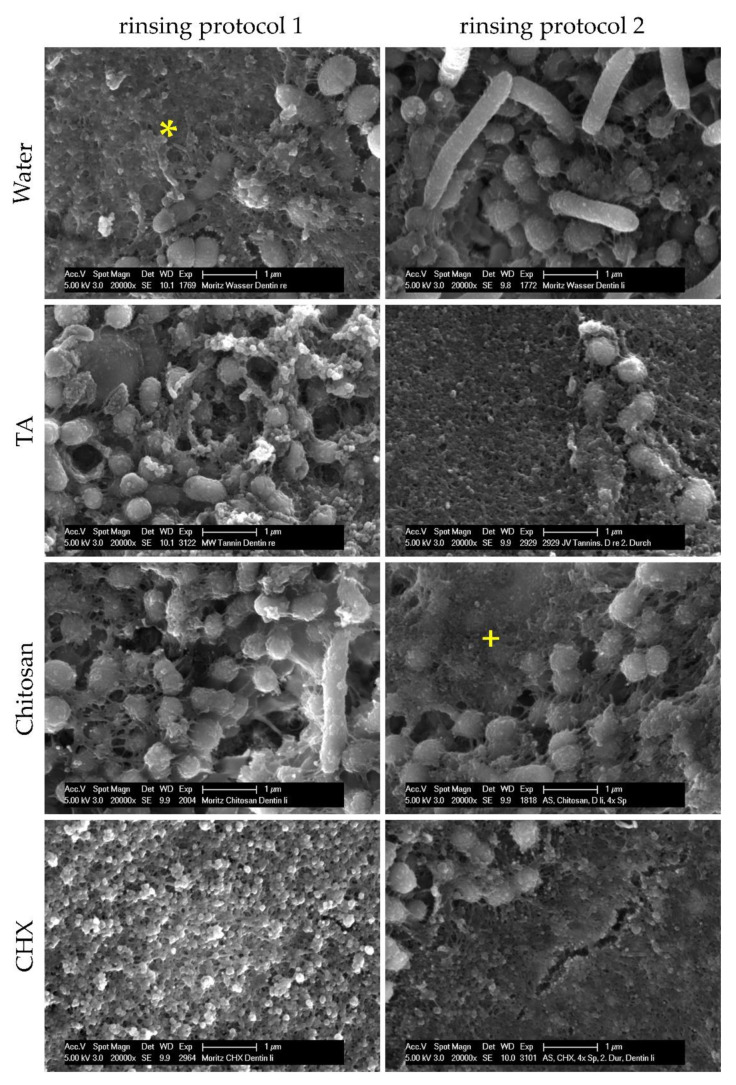
SEM images of dentin specimens at 20,000-fold magnification. Subjects carried intraoral splints with dentin specimens for 48 h and rinsed with different experimental solutions. In rinsing protocol 1, rinsing was performed after 3 min, 12 h, 24 h, 36 h and 47.5 h and in rinsing protocol 2 after 3 min, 12 h, 24 h and 36 h. Specimens were either covered by the pellicle (*) or by bacteria that were mostly cocci and a few rods. The bacteria were partially covered by the pellicle (+). Alterations of the pellicle structure occurred after rinsing with chlorhexidine.

**Figure 4 molecules-26-01351-f004:**
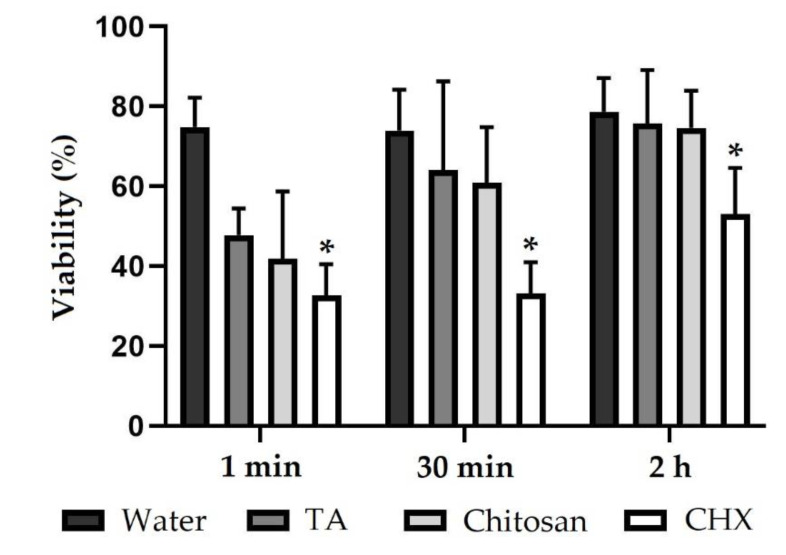
Viability (%) of bacteria in saliva samples. Subjects rinsed with different experimental solutions. Saliva samples were collected 1 min, 30 min and 2 h after rinsing. Saliva bacteria were stained with LIVE/DEAD^®^ Baclight™, and viability was evaluated using ImageJ software. The height of the bars corresponds to mean values and the line applied to standard deviations. Friedman test followed by Dunn’s multiple comparison test: significant differences (*p* < 0.05) to water are marked with *. TA = tannic acid, CHX = chlorhexidine.

**Figure 5 molecules-26-01351-f005:**
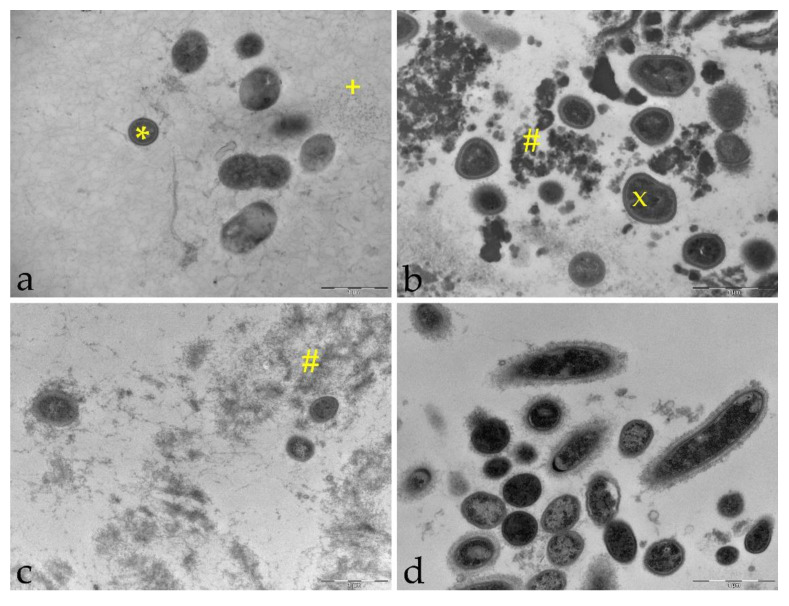
TEM images of saliva samples 1 min after rinsing with test substances. (**a**): negative control water, 18,500-fold magnification, (**b**): tannic acid, 23,000-fold magnification, (**c**): chitosan, 18.500-fold magnification, (**d**): positive control chlorhexidine, 23,000-fold magnification, intact morphology (*), loose protein-network (+), irregular morphology (x), protein aggregation (#).

**Figure 6 molecules-26-01351-f006:**
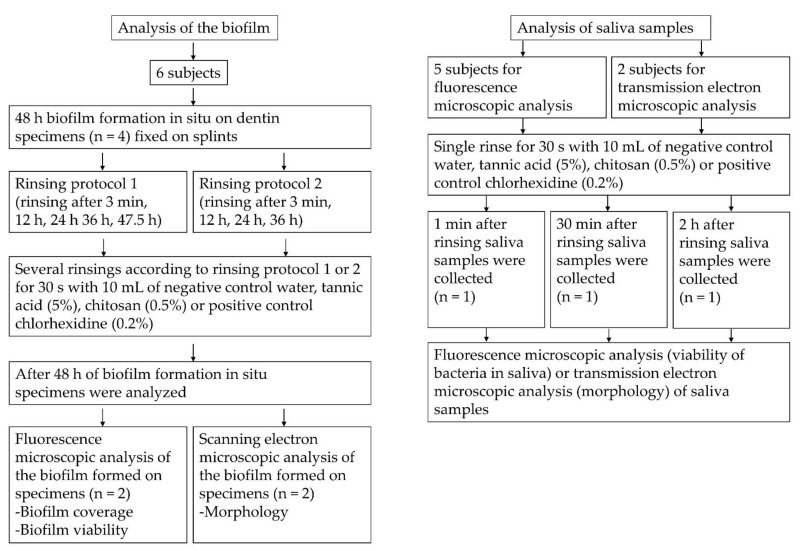
Flow chart of the in situ experiments.

**Table 1 molecules-26-01351-t001:** Modified scoring for biofilm coverage according to Xi et al. [25].

Score	Definition
1	Pellicle with no or scattered bacteria
2	Few and small bacterial aggregations, dozens of bacteria
3	Multiple bacterial aggregations, hundreds of bacteria
4	Monolayer biofilm or biofilm covering <50% of the surface
5	Multiple-layer biofilm covering >50% of the surface

**Table 2 molecules-26-01351-t002:** Scoring for biofilm viability according to Nobre et al. [62].

Score	Definition
1	Primarily red fluorescent bacteria, Ratio of red to green fluorescent bacteria is 90:10 or more
2	Ratio of red to green fluorescent bacteria is about 75:25
3	Ratio of red to green fluorescent bacteria is about 50:50
4	Ratio of red to green fluorescent bacteria is about 25:75
5	Primarily green fluorescent bacteria, ratio of red to green fluorescent bacteria 10:90 or lower

## Data Availability

The data presented in this study are available in the supplementary materials.

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
