# Peer review of "Evaluation of Anti-Biofilm Activity of Mouthrinses Containing Tannic Acid or Chitosan on Dentin In Situ"

_molecules, 2021, doi:10.3390/molecules26051351_

Round 1

Reviewer 1 Report

The problem of caries is very serious all over the world. It is important that we find the correct ingredients for the mouthwash. Such
ingredients that will have both anti-adherent and anti-bacterial properties,
as in the case of tannic acid.
It is also important that these components of
the mouthwashes should have the longest effect.
Thus the research presented
in the work I am reviewing is absolutely justified and touches upon a still
valid problem.
However, I have a question: why did the authors not use
longer rinsing times?
It is possible that this would affect on the results. Perhaps it is worth
doing such an experiment and seeing how this time will affect the
anti-adhesive and antibacterial abilities.

Reviewer 2 Report

This research is under the scope of this journal; the topic is relevant for readers, and this research deals with potentially significant knowledge to the field.

However, there are some concerns in the about the present manuscript:

Title

  • Evaluation on Dentin of Anti-Biofilm Activity in situ of two Mouthrinses Tannic Acid or Chitosan

Abstract

  • How many samples? Identified in the abstract.
  • The authors should describe how the results were expressed and statistical analysis performed. 
  • In the results, is important to show more information, add some of the p-values.

Keywords

  • Add more some keywords: Biofilm

Introduction

  • What is the importance of this study for clinical?
  • What is the gap in the literature? Is necessary to change some procedures on the dental office? Which results are comparable with others?
  • Page 2 line 45 – CHX have characteristic very important, it is a substantivity effect on human dentine. In endodontics was recommend was an intracanal medication, add (DOI:1016/j.joen.2013.10.023)
  • Page 2 line 72 - In the literature, the Chitosan have a different application in Dentistry – on of these applications was (DOI: 1016/j.joen.2017.03.005) investigated in an animal study the usage of chitosan scaffolds for dental pulp regeneration. This was added inside the root canal dentine walls to see recover of dental tissues. Please, add to support the dentistry application.
  • Page 2 line 75 - when you had in the text the “previously published study by Schestakow et Hannig (2020).” reference should come immediately afterwards, not in the final of the sentence. (corrected these in all manuscript).
  • The last paragraph used CHX abbreviation.
  • What was the null hypothesis for this study? Add on the last sentence in the introduction and reject in the discussion.

Materials and Methods

  • “0,2%” change for 0.2%. Correct this for numeric form in all manuscript.
  • ml change for mL
  • This section would better communicate with readers if restructured. A flowchart or diagram of the experimental processing would be valuable.
  • How was the sample calculated?

Did the authors perform power analysis to evaluate if this sample size was appropriate?

  • How many controls you have in this study? How many empty controls did you use?
  • How were the divided into groups? How was randomized the mouth rinses?
  • Did all subjects do four mouth rinses at the same time?… Clarified
  • Manufacturer’s instructions”! The manufacturer's instructions may be adjusted over time by the manufacturer himself. In your case, what instructions did you follow? (So, you should describe Supplementary material).

Results 

  • The font in the graphics is different from the text. And the resolution of the images can be improved.
  • Figure 3 and 5. Improved the identification of structures without numbers.
  • Figures - standardized the sized and the font in the figures and charts.  

Discussion 

  • Please, clarified what was the limitation of this study?
  • And also, clarified the future perspectives also add in the discussion.

References

  • But references are not standardized. The titles of references have a different format, 
    the title of the article is written in capital letters at the beginning of words, others only in lower case. Also, the standardized format of presentation in the journal's name. Because names have written in a different format, one is not abbreviated, others are not.
  • The reference is on the final of the sentence. But, when you had in the text the “authors et al.” references should come immediately afterwards, not in the final of the sentence. (corrected in all manuscript).

Reviewer 3 Report

Review of the article "Anti-Biofilm Activity of Mouthrinses containing Tannic Acid or Chitosan on Dentin in Situ". The task is clearly formulated, the prerequisites for the study are indicated, it is shown that tannic acids are a promising agent for rinsing teeth (it reduces the viability of bacteria and the adhesion of bacteria to dentin samples in situ). It has also been established that chitosan has an antibacterial effect only on non-sticking bacteria. Overall, the study is thoughtful and complete. I think there is only one question to be clarified. Why 48 hour biofilms are being investigated. It seems that all over the world, teeth brushing is done from one to several times a day. Also, the authors forgot to designate the country in which they work. 

Round 2

Reviewer 2 Report

This research is under the scope of this journal; the topic is interesting for readers and this research deals with potentially significant knowledge to the field and an open new way for future studies. 
The authors improved the quality of the manuscript after the reviewer's indications

Author Response

Dear Reviewer,

thank you very much for considering our manuscript for potential publication as a research article in the journal ‘molecules’. The manuscript was revised to address your comments. All the changes performed in the manuscript were highlighted by using ‘track changes’ function in Microsoft Word. After the corrections we hope that the manuscript meets the requirements and is suitable for publication in the journal ‘molecules’.

Thank you in advance,

sincerely

Anton Schestakow
